# "I've not been knocked out, so I'll probably be fine:" Amateur Rugby players do not know the risks they are taking

Daniel Walker[1]*, Adam Qureshi[2], David Marchant[3], Rebecca Murray[1],
Alex Bahrami Balani[2]

1 Department of Psychology, University of Bradford, Bradford, England, 2 Department of Psychology, Edge Hill University, Ormskirk, England, 3 Department of Sport and Physical Activity, Edge Hill University, Ormskirk, England

* d.walker5@bradford.ac.uk

## Abstract

Concussion is prevalent in British amateur Rugby and there is currently contradicting evidence of the attitudes and knowledge of concussion in Rugby players. As such this study aimed to investigate the reasons for the variance in attitudes and knowledge of concussion in British amateur Rugby. As concussion is a lived experience within sport, we utilised qualitative interviews using reflexive thematic analysis to assess data obtained from nine amateur Rugby players that had sustained over three concussions to understand if they are aware of the known risks associated with such injuries. Our findings show that *poor duty of care* from those around the player with suspected concussion was prevalent, and that this poor duty of care enables poor attitudes toward and limited knowledge of concussion. It was also discovered that a *lack of education* could explain this poor duty of care that is offered to Rugby players by non-players. *Poor duty of care* and *lack of education* encourage continued participation from the player with suspected concussion. When there are examples of World Rugby failing in their duty of care of the best players in the world, it is unsurprising that players and non-players in the amateur game follow suit. Therefore, the education of both players and non-players in amateur Rugby matches is paramount, as well as World Rugby improving their concussion practices, in tackling the poor attitudes and knowledge base that we see in the amateur Rugby game.

## Introduction

The dangers of concussion are becoming better-known within academia, and therefore it is necessary for this wealth of knowledge to be transferred to and applied in real life (i.e., sport activities). Rugby has been found to have the highest prevalence of concussions both in match play and training [1] when compared with other contact sports such as Association Football, Ice Hockey, and American Football. This is

**Data availability statement:** Transcripts are available at https://osf.io/3e6w5/

**Funding:** The author(s) received no specific funding for this work.

**Competing interests:** The authors have declared that no competing interests exist.

measured in Athletic Exposures (AE) which is defined as one athlete participating in one game or training session with the incidence rate shown as *x* per 1,000 AEs. Research has found Men's Rugby to have an estimated incidence rate of 12/1000 AEs in matches and 2.8/1000 AEs in training [2]. By comparison, Association Football players are found to have an incidence rate of 0.08/1000 AEs [3]. Although descriptive, these findings indicate that Rugby players may be at an increased risk of sustaining sport-related concussion and the negative consequences associated such as poorer mental health [4–6], impaired cognition [7,8], and increased risk for neurodegenerative diseases [9,10].

Despite the well-documented negative consequences associated with concussion, many Rugby players continue to play their sport, where concussion is common, and therefore placing themselves at risk of experiencing these negative outcomes. This could be indicative of poor knowledge and therefore poor attitudes towards concussion. For example, previous work has found that just 50.8% of Rugby players that had reported having had a concussion did so during the match or training session in which it occurred [11]. The most common reasons for non-reporting were not considering it a serious injury, not knowing that it was concussion at the time, and wanting to continue participating. These findings have been found previously in Irish professional Rugby Union players [12]. Unfortunately, this poor concussion knowledge often translates into poor concussion attitude, as 61% of Miller et al.'s [11] sample did not adhere to recommended return-to-play protocols after their most recent concussion sustained in Rugby.

It could be that Rugby players continue to play with suspected concussion so that they are perceived as "head strong". Colleagues have shown how players often downplay, deny, or conceal their concussion symptoms and play on, so they can present themselves as having a high level of commitment to their teammates and to their sport [13]. This has been reported in Men's and Women's Super League players where 20% of respondents stated that they did not report concussion symptoms during the 2020 and 2021 seasons [14]. 35% of the time this was due to concerns regarding team selection, where just under a quarter of the time (24%) this was due to feelings of not wanting to let the team down. These reasons are also found to be the main two motives for continued participation in the Australian National Rugby League [15] and in amateur Rugby in the United Kingdom [16]. Therefore, it could be that there is a culture where amateur Rugby players, despite their competitive status, often prioritise sporting values rather than health-related ones, rewarding serious risk taking [13]. This mentality can increase the chances of continued participation with a suspected concussion which can exacerbate symptomology [17]. Therefore, creating a team culture of reporting concussion symptoms is vital [18].

Concussion attitudes and knowledge base is variable within Rugby players with evidence that players possess safe [19] and unsafe [20] concussion attitudes. Viljoen et al. [21] also found poor knowledge in amateur Rugby players that may lead to poor attitudes and behaviour regarding concussion. Kraak et al. [19] assessed Western Province Rugby Union Super League players in South Africa using the Rosenbaum Concussion Knowledge and Attitude Survey (RoCKAS) [22]. Participants in this study

answered 67% of the Concussion Knowledge Index (CKI) questions correctly, and 62% of the Concussion Attitude Index (CAI) questions correctly with 65–70% considered a moderate level of knowledge of and attitude towards concussion [21,23]. Salmon et al. [24] also found that adolescent Rugby players in New Zealand and their coaches answered 69% and 74% of the CKI questions correctly, respectively. Coaches responded significantly better than adolescent players in this study, with the authors suggesting this was likely due to coaches being required to attend an annual RugbySmart course in New Zealand.

However, the use of RoCKAS is questioned for poor reliability [25] which is probably due to limiting participants' responses as they are restricted to predetermined questions and answers [26]. AlHashmi and Matthews [26] consider surveys a poor method of capturing complex human behaviour, such as attitudes towards and knowledge of concussion, suggesting that these can shift and change. Surveys only capture a snapshot of these and therefore it could be why we see variance in concussion knowledge and attitudes using RoCKAS. Van Vuuren et al. [20] report that players demonstrated the lowest knowledge of concussion and least safe attitudes towards concussion compared with medical staff and referees.

Aspers and Corte [27] define qualitative research as "…*an iterative process in which improved understanding to the scientific community is achieved by making new significant distinctions resulting from getting closer to the phenomenon studied.*" Qualitative research allows in depth insights into complex human phenomena, attitudes, perspectives, and behaviours. In contrast to quantitative research, the nuances which underpin lived experiences are illuminated within qualitative research. Such lived experiences are central to impactful educational research which is invested in the hows and the whys which are not represented easily by numbers [28]. According to Allen et al. [29] qualitative public health research can provide profound understandings of beliefs and behaviours which can feed into problem solving public health messages.

For example, Seguin and Culver [30] collected qualitative data across a two-year period that included two interviews, five focus groups, 24 member reflections, and 31 informal discussions that uncovered a unique understanding of lived experience through Bronfenbrenner's Ecological Systems Theory [31]. They concluded that, i) microsystem – athletic identity, ii) mesosystem - (dis)trust in relationships, iii) exosystem – concussion protocols, iv) macrosystem – sport culture, v) chronosystem – timing related to major events and recovery. This work can be credited for providing a unique perspective regarding the lived experience of concussion in sport albeit not explicitly specific to Rugby, however Rugby players were included. In addition, Daly et al. [32] interviewed 23 retired professional Rugby players and identified that the players did not fully understand the consequences of concussion because it was normalised within the game. Dismissive language was commonplace which supports previous research of not reporting concussion due to not considering it a serious injury [11,12]. Therefore, a qualitative approach via interviews to explore the knowledge and attitudes of Rugby players towards concussion is optimal and can uncover nuance that RoCKAS may be unable to.

In summary, the literature using RoCKAS [22] to assess the knowledge and attitudes of concussion in Rugby players is inconsistent, and qualitative approaches can provide profound understandings into lived experience [29]. Therefore, qualitative interviews should be conducted to investigate this lived experience. The notion that Rugby players understand the risk of concussion yet continue to place themselves at risk of sustaining one is of particular interest. Therefore, this study aimed to interview amateur Rugby players to investigate their attitudes and knowledge of concussion in their sport and provide indications as to how we can best support players.

## Method

### Research paradigm

The researchers adopted an interpretivist ontological view when constructing the research question as the interpretivist paradigm allows subjective experiences to be explored here within the social context of sport. Thus, the world is complex and dynamic, and people experience reality in different ways [33]. As concussion is becoming more known as a personal

experience [34], it was believed that an interpretivist view was appropriate, to uncover the subjective reality of the participants that have undergone this head trauma in sport. It is important to understand how some sportspeople develop poor attitudes towards concussion that could lead to the negative consequences associated with continued participation.

## Situating the authors

The authors have not sustained concussion first-hand but take a keen interest in sport which has made the increasing attention on head injuries, particularly in contact sports, unavoidable. When leading healthy, balanced, active lifestyles, the authors are motivated for others to adopt a similar outlook. We have good knowledge of concussion having researched this area, as well as contextualising this in a sporting setting. We took an ontological view, with the belief that people experience reality in different ways. Therefore, we implemented a semi-structured design that allowed us to ask follow-up questions during interviews to explore the way in which sportspeople reach their knowledge base of and attitudes towards concussion.

## Participants

Nine amateur Rugby players (7 Male, 2 Female) were included in the sample, all of whom were English (Age, $M = 24.67$, $SD = 5.15$). All players had sustained a minimum of three concussions during their playing careers. Amateur status was defined by the sportspeople themselves and therefore is operationalised as self-identification being an amateur Rugby player. Further information regarding concussion profiling and participant characteristics are presented in Tables 1 and 2, respectively. Participants were recruited via advertisements on social media platforms such as X and LinkedIn as well as word of mouth. All participants took part in semi-structured interviews designed to investigate their attitudes and knowledge towards concussion in Rugby. Interviews were conducted between January 2021 and March 2022.

## Procedure

Upon agreeing to take part, participants were invited by email to an individual Zoom meeting with Dr Walker that lasted 45–60 minutes. Semi-structured interviews were utilised which began with a set of "warm-up questions." These questions included details about their sporting history (e.g., when they began playing, what sports they had played, what competitive levels etc.), their preference between team and individual sports, how competitive they are before then asking about their experiences with concussion. Following these questions, it was of interest to learn about how much the players knew about concussion, their interest on learning more, whether they are aware of the dangers of concussion, continuing participation with a suspected concussion, and their opinions on adhering to recovery protocols following concussion.

## Ethics

British Psychological Society (BPS) ethical guidelines were adhered to with data collection commencing after ethical approval was obtained from the university. A participant information sheet informed participants of the nature of the study and their rights as a participant including details on the withdrawal of data if they wished to do so. All participants were 18

**Table 1. Concussion Profiles.**

|  | *N* |
|---|---|
| Total Participants | 9 |
| Sustained over three concussions | 9/9 (*Range* = 3–20, *M* = 8.13, *SD* = 5.59) |
| Sustained over five concussions | 5/9 (*Range* = 3–20, *M* = 8.13, *SD* = 5.59) |
| Sustained first concussion before 18 years of age | 5/9 |
| Continued participation with concussion symptoms | 9/9 |
| Witnessed others continue with concussion symptoms | 9/9 |

**Table 2. Participant Characteristics.**

| Participant | Age at the time of interview | Years since last concussion at the time of the interview | Playing information at the time of interview | Competitive levels played at the time of interview |
|---|---|---|---|---|
| Chloe | 20 years old | Less than 1 year | Has played Rugby for the past 8 years and continues to do so. | Has played for community and university teams. |
| Harry | 20 years old | 3 years | Has played Rugby for the past 9 years and continues to do so. | Has played for community and university teams. |
| James | 22 years old | 2 years | Played for a total of 16 years between the ages of 5 and 21. Retired last year. | Has played at academy level and BUCS Super Rugby. |
| Joshua | 33 years old | 1 year | Played for a total of 17 years between the ages of 15 and 32. Retired last year. | Has played for community clubs. |
| Luke | 26 years old | Less than 1 year | Has played Rugby for the past 19 years and continues to do so. | Has played for community and university teams. |
| Matt | 22 years old | 1 year | Has played Rugby for the past 14 years and continues to do so. | Has played for clubs in England, France, and Wales. |
| Ryan | 21 years old | 1 year | Has played Rugby for the past 17 years and continues to do so. | Has played at academy level and for university teams. |
| Sophie | 33 years old | 2 years | Has played Rugby for the past 12 years and continues to do so. | Has played for community and university teams. |
| Tom | 25 years old | 7 years | Played for a total of 10 years between the ages of 8 and 18. Retired 7 years ago. | Has played for community clubs. |

years or older at the time of completing the study. After the interview, a debrief form was emailed to them reiterating the aims of the study and reminding participants how to withdraw their data if they wished to do so.

## Data analysis

Thematic analysis is a widely used technique in qualitative sport and exercise research [35,36] and therefore was adopted in the present study. Given that Braun and Clarke have been consistently concerned with the usage of their approach since their initial paper was published, we followed the six-phase process that the authors proposed [37]. Although these steps are presented in a logical sequential order, this is not a linear process and therefore we moved back and forth through these steps during analysis. In addition to the six stages, we also utilised the 15-point checklist recommended by Braun and Clarke [38]. This allowed us to align our analysis with a rigorous approach. The analysis therefore went beyond the six stages to a more complex method which substantiated the validity of the conclusions drawn and presented here.

Following this thorough approach to analysis, two themes were generated which captured the recurrent patterns across the data set: 1) Poor duty of care, and 2) Lack of education. We turn to these findings now.

## Findings

### Poor duty of care

Participants recalled examples where poor duty of care was evident from various sources in their playing careers. Many suggested that their coaches possess unsafe attitudes towards concussion, perhaps inherited from the coaching style they received in their own playing career [39]. It is unlikely that these coaches are actively attempting to cause harm to their players, instead more likely trying to "toughen them up" and to become "head strong" [13]. However, research on group dynamics and behaviour suggests that individuals will not only observe the social rules within the group but also outside of it to secure their place in the group [40] and therefore this poor practice encourages players to adopt the same attitudes.

Matt stated "sometimes…your manager…will be a bit understanding but…if you're a forward (the) forwards coach will be…like you're getting back on…and a lot of your teammates will try and like hype you up…you're alright…and I'm

probably guilty of that. I've probably done that as well." Tom also claimed "the doctor basically said like you'll be fine. Get back on and I don't remember playing the rest of the game." This is a startling revelation given that the club doctors are supposedly there to ensure safety, but Matt goes on to say that they had known "a lot of people who've been like you need to go back on" highlighting a culture of poor duty of care in amateur Rugby. This is like what Sophie reported when she stated she had been "splashed with water and told to carry on" when she suspected she had concussion herself and Ryan claiming he had "even seen a bloke be knocked out and the coach didn't pull him off and he just carried on playing."

Tom even spoke of the referee applying pressure and a poor duty of care when they had a suspected concussion "the referee had stopped the game because we were playing with fewer numbers anyway and said, well, is he coming back on because it was just a nosebleed in his eyes. He was happy to stop the game and wait for me to come back on which put an inordinate amount of pressure (on me)." Unfortunately for this player, the referee had stopped the game and now there was more pressure to continue from his parent "my dad said go on get back out there and I sort of blindly ran out and as a 14-year-old you didn't really think anything of it." Here is an unfortunate example of poor duty of care from match official and parent, where with better concussion knowledge and understanding this player could have been protected from the negative consequences and risk associated with continued participation with suspected concussion. Importantly, this player is frustrated at the lack of care from those around them "if he hadn't stopped at that juncture for me to come back on, I probably wouldn't have come back on because I would have had the adrenaline come down and go, actually, why am I going back on my nose hurts?"

These passages highlight poor duty of care from coaches, match officials, and parents that are then adopted by players. External pressure is increasing the number of amateur Rugby players that continue playing with potential concussion and placing themselves and others at exceptional risk. Therefore, improving the knowledge base of coaches, match officials and parents in the amateur game will in-turn improve the duty of care they feel towards their players. A *lack of education* was identified as a theme, where the following passages will highlight this.

## Lack of education

Harry stated, "most of it (concussion education) was just from like the posters that you see like around the Rugby club that was about it." This was corroborated by Tom who said "the first one was because I didn't know what was going on. In terms of...I've never heard of concussion. I never thought about the fact that my headache or the symptoms I was feeling was to do with head injury." What is more worrying is that Tom finished their point with "what probably prompted me to play on more so was the pressure put on me." Luke said that they had never "had a session where we sat down and said this is what concussion can do to you, this is what the long-term effects are and that sort of thing."

This lack of education is found elsewhere in the transcripts where Luke displays poor knowledge of concussion symptoms, "I've not been knocked out, so I'll probably be fine" which is also seen in Joshua "I had read things on that (RFU Headcase course), and I was thinking, I didn't realise that constituted concussion. So I was thinking, Oh, well, I hadn't had concussion before that, but then you read these symptoms and signs. And it's like, oh bloody hell maybe I actually have?" This poor knowledge often translates to poor attitudes such as James stating that "(you experience) a little bang and nobody's really seen it and the doc hasn't pulled you off and you feel alright, you can see things and your head's sort of alright, you just carry on don't you?"

These passages demonstrate a lack of education whereby players state they were not aware that the symptoms they were experiencing were due to concussion. Players also state that they felt pressure to continue which highlights the poor concussion attitudes from players and non-players. An issue that is present is that any gap in knowledge seems to be filled through experience, where those that know the most about concussion are the ones that have sustained one.

Luke stated "I would say most of our squad does (understand concussion) but again probably because they've had one. New players might not be quite as aware of what's going to happen." They also made the point that they had had physios try to explain concussion to them at their club but that "apart from that it's mainly just been from, you know like,

seeing it on the news and hearing about, hearing about players that are not feeling well, not doing too great after, after it's done." Ryan also reported that they were not aware of what concussion was until experiencing it first-hand "I'd smash my head, have all the symptoms but didn't really know what it was until I got diagnosed for the first time." Another concern is Sophie stating that "if you get the symptoms straight away it's different, but you can get concussed and feel a bit iffy. But it's not visible. So our coach will say oh you're fine, carry on."

These findings of poor duty of care and lack of education help to explain what we aimed to investigate in this study, which was to uncover the reasons for players taking the risk to play Rugby, when there is a high chance of ill-health associated with it. Here, it has been reported that generally, amateur Rugby players in the United Kingdom possess a low level of understanding of concussion, until they sustain one. It is in our view, that prevention is the best treatment of concussion, but at present Rugby players are making uninformed decisions on their participation. This is further compounded by a culture of a poor duty of care in Rugby. Our findings seem to suggest that teammates, parents, coaches, and even officials display a poor duty of care, and even pressure players to continue with suspected concussion in some instances.

## Discussion

This study attempted to explore the knowledge and attitudes of concussion in amateur Rugby. This has been deemed necessary due to the wide-ranging knowledge bases [41] and positive and negative attitudes towards concussion in Rugby [19–21]. Many Rugby players report that they have good attitudes and appropriate knowledge of concussion, yet this seems contradictory. For one to state that they are aware of potential neurodegenerative diseases and to choose to continue participating in Rugby is indicative of the difficulty of changing health-related behaviour [42]. Kelly and Barker [42] argue that there are six common errors when discussing health-related behaviour change, i) It's just common sense, ii) It's about getting the message across, iii) Knowledge and information drive behaviour, iv) People act rationally, v) People act irrationally, vi) It is possible to predict accurately, and these six errors may be evident when conveying and interpreting the risks associated with concussion in Rugby. For example, Tadmor et al. [14] provide support for the fifth common error that Kelly and Barker [42] propose in that people behave irrationally in their response to concussion in Rugby. They found that 11% of their sample of 422 Men's and Women's Rugby League players would dissuade family members' children from playing Rugby League but continue to do so themselves. This behaviour is illogical as it places the health of others above one's own, as well as demonstrating an understanding of the risks associated with concussion, and actively opting to ignore them. Therefore, this study aimed to investigate and uncover the reasons as to why Rugby players continue to play their sport, despite the well-reported potential negative consequences.

From our data analysis, we developed two themes, *poor duty of care* and *lack of education.* It is unsurprising to find that there is a poor duty of care found within amateur Rugby given this has been a topic of discussion for some time [43,44]. One reason for this may be a bystander effect whereby the presence of several people in an injury situation, such as concussion, may lead to a reduction in an individual stepping in to help [45]. Salmon et al. [46] identified 30 distinct responsibilities related to concussion management in amateur Rugby and that there was a distinct lack of clarity regarding personnel accountable for each. Therefore, role confusion could aid in explaining poor duty of care from non-players in amateur Rugby.

Additionally, retired professional Rugby Union players across Ireland, England, Scotland, and Australia report that they believe owners of Rugby clubs should have a duty of care towards their players during retirement as well as during their playing careers [47]. In a way, as this is a topic of discussion, these sentiments demonstrate that former professional Rugby players feel as though they were not supported when playing and in retirement. While unreasonable, the motive for professional Rugby clubs to distance themselves from former players that then go on to develop negative outcomes associated with concussion is understandable. At the time of writing, Rugby is under scrutiny regarding the negative consequences of concussion and governing bodies are currently under litigation in the United Kingdom [48]. World Rugby annually provide grant funding for player welfare research, but it is unclear how many awards are made, the cost of these

awards, and how much money goes towards concussion research [49]. This makes it difficult to understand whether governing bodies are supporting players in retirement appropriately as is reported by Daly et al. [47].

With that said, it may be that there is a role confusion as to who is responsible for supporting retired Rugby players the same way there is for who supports the injured player in that moment. For example, World Rugby may suggest it is the responsibility of the Rugby Football Union (RFU), who may say the opposite or point towards different charities for support. Ultimately, this lack of clarity can result in Rugby players not being appropriately supported directly after a concussion, or when suffering the consequences years later in retirement.

This is where the second theme of lack of education emerges. It is unlikely that anyone involved in a Rugby match, at any competitive level would like to see a player injured, particularly a brain injury. Therefore, our interpretation of why we see a poor duty of care from teammates, parents, coaches, and referees, is due to a lack of education on concussion. This behaviour is indicative of poor knowledge of concussion, rather than the alternative that they are willingly turning a blind eye to potential significant brain injury of their teammate, child, or player they are coaching or officiating.

This lack of education and poor knowledge of concussion is consistent with the literature. Kearney and See [50] found that in their sample of 255 English youth players aged 11–17, that 61 reported having had a concussion, a prevalence of 24%. Of those 61, only 7 (11%) of them followed return to play protocol, with some not adhering to it at all. In fairness, players also did show some good knowledge of concussion with 80% correctly identifying that a concussion does not have to result from a hit to the head, and 91% correctly identifying that those that have sustained a concussion in the past are not less likely to have another. Despite this positive knowledge, this does not always transfer to positive attitude, as we see that 36% reported that they would continue playing despite hitting their head and 30% agreed that they had a responsibility to continue playing with concussion symptoms. Unfortunately, 24% reported that their parents would encourage them to continue playing in a Rugby match after hitting their head, indicative of external pressures influencing this decision.

Our study builds on these findings from Kearney and See [50] supporting the narrative that Rugby players often continue with concussion symptoms and that this is in-part due to a poor duty of care from non-players. The Coaches' code of conduct set out by World Rugby [51] states that coaches are *"...responsible for maximising benefits and minimising risks to players"* and that they *"...should attain a high level of competence through qualifications and a commitment to ongoing training that ensures safe, current, and correct practice."* World Rugby [52] also state that *"Match officials ensure the players' safety and ensure that the players follow the principles of the Game. Match officials might need to stop the game to ensure player safety."* The interview transcripts in this study highlight that both coaches and referees have failed in their responsibilities set out by World Rugby which we deem to be due to a lack of education. This is concerning given that there is evidence that medical staff and referees in South Africa demonstrate highest concussion knowledge and safest attitudes compared to players [20]. However, these findings may only be representative of South African medical staff and referees as it may be that training and discussion towards concussion in South Africa exceeds that of what is available in the United Kingdom. For example, South African referees can exercise a Blue Card which removes players with suspected concussion from the match [53]. This has also been implemented in Australia [54], Canada [55], and New Zealand [56] whereas this has not yet been implemented in the United Kingdom. As this is not present in the United Kingdom, there could be increased pressure on continuing participation rather than having this decision taken away from the player due to being shown a Blue Card by the referee.

As we have discussed in previous work [6], quite often the reluctance towards concussion information is due to the lack of a clear message. Players sometimes oppose the message that concussion is a significant public health issue as there can be no causal link established, and that any research shows an association, or a possibility of developing negative post-concussion outcomes. However, Nowinski et al. [57] have suggested that a causal relationship between repetitive head impacts and chronic traumatic encephalopathy can be established when evaluating relevant studies using Bradford Hill criteria. These findings have also been supported by establishing that sustaining a concussion exposes sportspeople

to a 57 times greater risk of developing meaningful depressive symptoms [6]. This type of work may be useful in changing the knowledge and attitudes of non-players in order to promote a healthier sense of duty of care towards Rugby players.

The simple message that Walker et al. [6] provide is in comparison to the comprehensive guidance currently distributed by World Rugby and the RFU. World Rugby provide a ten-page concussion guidance document [58] that outlines what concussion is, symptomology, and the correct return-to-play protocol for adults and children. The RFU has its own Head-case programme [59] that aims to increase the understanding and provide information on concussion. These documents are accessible to Rugby clubs in England and around the world but whether players and non-players actively engage with them is unknown. The findings of the present study would suggest that these documents are rarely accessed by amateur Rugby players in the United Kingdom. This guidance might create the illusion that World Rugby are dedicated to concussion prevention within their sport, but there is a case to be made of actions speaking louder than words. In 2023, France hosted the Rugby World Cup and French captain Antoine Dupont was substituted after a head collision in a Pool match against Namibia. Although Dupont did not play in France's final Pool match against Italy, he did start their Quarter-Final match against South Africa. This came despite media attention that Dupont was still suffering from concussion symptoms.

While World Rugby state in their concussion guidance that *"if in doubt, sit them out"* and that *"any player with a concussion or suspected concussion should be immediately and permanently removed from training or play"* allowing Dupont to compete in a Quarter-Final of a World Cup, sends the opposite message. This is an example where World Rugby have ignored their own guidance and supports critique of using the "*if in doubt, sit them out*" slogan on its own as it provides governing bodies in sport the chance to create the illusion that ongoing efforts are being made regarding concussion prevention and deflects criticism from their sports [60]. This decision also opposes multiple components of the Health Belief Model [61] which has been found to be significantly related to positive concussion reporting intentions [62]. If the world governing body of Rugby are contradictory in their concussion behaviour and duty of care towards players, this could reduce the perceived susceptibility of concussion for Rugby players of any competitive level. They may feel less vulnerable to the disease as they observe one of the best players in the world bounce back to competing in a World Cup Quarter-Final and feel encouraged to emulate this behaviour. The perceived severity of concussion may also be reduced which in-turn could hinder the cue to action and discouraging behaviour change. It is also important to note that World Rugby have not represented concussion in their sport fairly in the past and have in-fact previously downplayed and denied the long-term risks [63]. This undermines the possibility for anyone to be educated appropriately on concussion in Rugby. If players and non-players in Rugby are going to protect those with concussion, it must start from the top as at present the actions of World Rugby demotes key components of the Health Belief Model which have been found to be associated with positive concussion attitudes [62].

## Limitations

Although other studies that have investigated this area have obtained larger samples [13,26], qualitative research is less concerned with sample sizes than typical quantitative approaches. Instead, the depth and richness of the Rugby players' experiences of concussion were of interest here. This was successfully explored, as all nine participants had sustained over three concussions with one having sustained an estimated twenty. All players admitted to continuing playing with concussion symptoms as well as seeing others do the same. Over half had sustained concussion before adulthood. Therefore, the sample are well-qualified to share their lived experiences of concussion, and the findings here are valuable.

Our recruitment strategy, however, should be noted. We recruited participants using advertisements across the university as well as with social media posts, and therefore the data may suffer from self-selection bias. The findings that we have obtained may reflect individuals that have had the worst experiences post-concussion. Many sustain concussion and show no symptoms [64], and therefore these players may have less motivation to take part in a study attempting to understand concussion in amateur Rugby like the present study. By contrast, those that have dealt with more negative experiences post-concussion are perhaps more likely to take part to help others and that could explain the current findings. We

would argue that even if this is the case, there is value in examining the worst experiences of concussion satisfying the ontological view of the researchers. These nine amateur Rugby players highlight the negative consequences of concussion in their sport and their contributions can promote positive concussion attitudes and knowledge.

The reader should also be aware that these interviews were retrospective, self-reported accounts. As Table 2. illustrates, the experiences that were being recalled ranged from less than a year to seven years before the interview took place. However, the interviews captured data from various viewpoints, not just their own. Questions were asked regarding observing others sustaining concussion, as well as their thoughts regarding how others would perceive a certain situation, such as their coaches or their teammates. Therefore, although the concussion they sustained may have been some time prior to interview, valuable information was still captured regarding their attitudes and knowledge of concussion regardless of this event.

Additionally, participants all competed in amateur Rugby in the United Kingdom, limiting these findings to just this demographic of players. This is considerably different to the way concussion in the amateur Rugby game is reported and managed in other popular Rugby playing nations such as Australia, Canada, New Zealand and South Africa where referees can use a Blue Card to remove a player from the game that is suspected to have sustained a concussion [53–56]. Therefore, the findings of the present study can only be applied to nations where the Blue Card system is not in use as it reduces the possibility of continued participation for the player.

## Conclusion and future directions

In conclusion, this study provides a deeper understanding of the variable attitudes and knowledge base of amateur Rugby players towards concussion in the United Kingdom. A poor duty of care coupled with a lack of education promotes poor attitudes and knowledge of concussion that we see in amateur Rugby players. This in-turn leads to continued participation with suspected concussion which is dangerous [17]. We therefore conclude that our sample of amateur Rugby players in the United Kingdom represent a cohort that may not be appropriately educated on the risks of concussion and not suitably cared for by non-players. Improving education of players and non-players in amateur Rugby could enhance the duty of care towards players as well as improving the chances of players making an informed decision on their participation. Finally, Rugby governing bodies like World Rugby and the RFU must set the example regarding duty of care. At present, there are instances where the two fail in their duty of care of current and former players which needs improvement if this is to be observed in the amateur game.

## Author contributions

**Conceptualization:** Daniel Walker.

**Data curation:** Daniel Walker.

**Formal analysis:** Daniel Walker, Rebecca Murray.

**Investigation:** Daniel Walker, Rebecca Murray.

**Methodology:** Daniel Walker.

**Project administration:** Daniel Walker.

**Resources:** Daniel Walker.

**Software:** Daniel Walker.

**Supervision:** Adam Qureshi, David Marchant, Alex Bahrami Balani.

**Validation:** Adam Qureshi, David Marchant, Alex Bahrami Balani.

**Writing – original draft:** Daniel Walker.

**Writing – review & editing:** Daniel Walker.

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
