## [Decision Letter · Decision Letter 0]

4 Mar 2025

PONE-D-24-58092“I’ve not been knocked out, so I’ll probably be fine.” Amateur Rugby players do not know the risks they are taking: This needs to changePLOS ONE

Dear Dr. Walker,

Thank you for submitting your manuscript to PLOS ONE. After careful consideration, we feel that it has merit but does not fully meet PLOS ONE’s publication criteria as it currently stands. Therefore, we invite you to submit a revised version of the manuscript that addresses the points raised during the review process. Please refer to the reviewer's comments for details, but the key areas to address are concerning:

More explicit statements of the knowledge gaps being filled, and the practical applications of the data.Relating the study explicitly and rigorously to previous and related work on this topic.Discussion of how generalisable the data are given the specific sample population.Issues regarding the availability of data in a way that might prevent anonymity.More discussion around the rigor of the methods and analyses used.

We look forward to receiving your revised manuscript.

Kind regards,

Christopher Kirk, PhD

Academic Editor

PLOS ONE

Journal Requirements:

2. Please ensure that you include a title page within your main document. You should list all authors and all affiliations as per our author instructions and clearly indicate the corresponding author.

Reviewers' comments:

Reviewer's Responses to Questions

**Comments to the Author**

1. Is the manuscript technically sound, and do the data support the conclusions?

Reviewer #1: Yes

Reviewer #2: Partly

2. Has the statistical analysis been performed appropriately and rigorously? 

Reviewer #1: N/A

Reviewer #2: I Don't Know

3. Have the authors made all data underlying the findings in their manuscript fully available?

Reviewer #1: Yes

Reviewer #2: Yes

4. Is the manuscript presented in an intelligible fashion and written in standard English?

Reviewer #1: Yes

Reviewer #2: Yes

5. Review Comments to the Author

Reviewer #1: Dear authors, thank you for the opportunity to review your manuscript, please find my recommendations below.

Overall Assessment

This manuscript addresses an important and timely issue in sports medicine and concussion awareness in amateur rugby. The use of qualitative interviews is an appropriate method to explore the attitudes and knowledge of players regarding concussion risks. The manuscript is generally well-structured and presents clear findings. However, there are several areas where improvements can be made, particularly in contextualizing the results within the existing literature, refining the discussion, and addressing ethical concerns regarding data accessibility.

Detailed Review

1. Title and Abstract

Strengths:

The title is engaging and effectively captures the study’s focus.

The abstract concisely summarizes the study’s aims, methods, and key findings.

Areas for Improvement:

The study’s objective should be more explicitly stated in the abstract. It is currently implied that the study investigates a lack of knowledge, but this could be clearer.

Consider including a more direct mention of the study’s practical implications (e.g., policy, education, or intervention strategies).

2. Introduction

Strengths:

The introduction effectively situates the study within the broader context of concussion risks in rugby.

The motivation for the study is clear.

Areas for Improvement:

Although the authors have incorporated additional recent studies after previous review comments, the knowledge gaps that this study aims to address should be more explicitly stated.

The discussion of concussion incidence in rugby compared to other sports (football, American football) could be further elaborated.

While recent studies (e.g., Tadmor et al., 2023; Longworth et al., 2021) have been integrated, it would be helpful to explicitly state what gaps remain in the literature.

3. Methodology

Strengths:

The use of reflexive thematic analysis is appropriate for the research question.

The study provides transparency by outlining the authors’ positioning in the research.

Areas for Improvement:

The sample size (n=9) is small and limited to England. While qualitative research does not require large samples, a brief discussion of how this affects generalizability could be included in the methods section.

The rationale for selecting participants with at least three concussions should be explained further. Could including players with fewer concussions provide additional insight?

The authors discuss their own positionality, which is commendable, but potential biases in data collection and interpretation should also be acknowledged.

4. Results

Strengths:

The two main themes ("Poor Duty of Care" and "Lack of Education") are well-developed and supported by participant quotes.

The authors provide a strong narrative illustrating players’ lived experiences.

Areas for Improvement:

Consider exploring subthemes within the two overarching themes to provide more nuance.

Were there any differences in responses between male and female participants? If so, these should be discussed.

A visual representation (e.g., thematic map or summary table) could help clarify the findings.

5. Discussion

Strengths:

The study is well-integrated with existing literature.

The discussion of policy implications and the role of governing bodies is relevant.

Areas for Improvement:

Some sections of the discussion read more like results rather than analysis. A deeper interpretation of findings in relation to past studies is needed.

The study’s unique contribution compared to previous qualitative research should be more explicitly stated.

The discussion about RFU and World Rugby concussion education programs is useful, but specific suggestions for improvement should be provided.

6. Ethical Concerns Regarding Data Accessibility

One major concern is that all interview transcripts are openly available online. While transparency is valuable, this raises significant ethical and data protection issues:

Confidentiality risks: Even if anonymized, participants' experiences might be recognizable to teammates, coaches, or clubs, leading to potential social consequences.

GDPR and ethical compliance: Under data protection regulations (e.g., GDPR), even indirectly identifiable information (e.g., age, gender, club affiliations) could pose privacy risks.

Risk of data misuse: Open-access qualitative data can be taken out of context or used in ways that the researchers did not anticipate.

Recommendation:

Restrict access to transcripts through a controlled repository, requiring researchers to apply for access.

Ensure full anonymization before data sharing.

Clarify in the manuscript why the transcripts were made publicly available and how participant confidentiality was ensured.

This ethical concern should be addressed before publication.

7. Limitations and Future Research

Strengths:

The study acknowledges key limitations, including sample size and demographic constraints.

The suggestion for future research to include a broader sample is appropriate.

Areas for Improvement:

Consider recommending longitudinal studies to track whether educational interventions lead to lasting changes in concussion attitudes.

A brief discussion of cultural differences in concussion management between different rugby nations (UK, South Africa, New Zealand) could strengthen the study’s conclusions.

8. References

Strengths:

Most key studies are cited.

Additional references have been incorporated after the previous review.

Areas for Improvement:

Ensure all cited studies are listed in the reference section (e.g., Walker et al., 2023 was missing previously).

Incorporate additional recent studies (2023–2024) where relevant.

Final Recommendation: Minor Revisions Required

This study makes an important contribution to concussion research in amateur rugby. However, minor revisions are necessary before publication. The most critical issue is the open accessibility of interview transcripts, which must be addressed to comply with ethical and data protection standards. Additionally, clarifications in the methodology, discussion, and references would strengthen the paper.

Summary of Key Revisions Needed

✅ Clarify the study’s objective in the abstract.

✅ Better define knowledge gaps in the introduction.

✅ Discuss sample selection and generalizability in the methodology.

✅ Provide deeper analysis in the discussion (e.g., gender differences, policy implications).

✅ Consider adding a thematic map or summary table.

✅ Address serious ethical concerns regarding open access to interview transcripts.

✅ Ensure all references are correctly listed and updated.

Once these issues are addressed, the manuscript will be in a strong position for publication.Overall Assessment

This manuscript addresses an important and timely issue in sports medicine and concussion awareness in amateur rugby. The use of qualitative interviews is an appropriate method to explore the attitudes and knowledge of players regarding concussion risks. The manuscript is generally well-structured and presents clear findings. However, there are several areas where improvements can be made, particularly in contextualizing the results within the existing literature, refining the discussion, and addressing ethical concerns regarding data accessibility.

Reviewer #2: This paper discusses a pertinent issue in sport, however, there is a need to further discuss the findings and methods in more detail. Some content within this paper is written in sensationalist/colloquial language, which should be toned down for publication in the scientific context. A focus on “referenceable ” sources should be maintained and more detail is needed within the methods and results. Please find some comments below that I hope will support the development of this work.

Introduction

Line 60 -62 arguably in the Miller example there is no reference to the reasons behind the non-reporting. I think this sentence needs a link between them , and into the Fraas reference on line 66. Again on line 67 any assessment of knowledge/ its relationship with behaviour in the Miller ref is not discussed.

From line 86 concussion education/ knowledge is discussed, however I feel this section would benefit from discussion of more recent literature – the world rugby team (Danielle salmon etc) have published some relevant work on this.

In the introduction, there is little discussion of the existing literature that reports lived experience of concussion education and how it may influence experience, despite this being the focus of the paper. I feel this section would be strengthened by greater discussion of this, and the point that survey’s only capture a snapshot should be made stronger. In line 106, I feel the aim should be flipped so that the value of qualitative investigation to capture the rich/complexity and transiency of concussion symptoms/ lived experience, is highlighted, rather than a more vague reference to the RoCKAS as inconsistent. This paper adds value to the often numbers based literature on this topic, and its important to highlight that!

Method

Line 117 please explain this point further, at present I am not sure what this means – highly individual perhaps?

In the situating the authors paragraph 122, positionality could be discussed in greater depth. The authors are described, but how did the author position influence interview data? The phrase “the authors are motivated for others to adopt a similar outlook” almost implies the authors have an agenda, or may ask leading questions. I’d suggest rewording this more neutrally, or detailing explicitly how this motivation may specifically influence the interviews. Do the authors have rugby playing experience?

If this data were collected, it would be more relevant to report ps gender, rather than sex, given the gendered experiences of concussion referenced within existing literature and discussion of social, rather than biological phenomena. If you don’t have this data, a sentence added to explain this would be beneficial. Please discuss how/why a sample of 9 participants was used – was there any screening, how many people expressed interest in taking part?

Please could you discuss the topics discussed during the interview in more detail and make the interview guide publicly available. Are all the interview topics relevant to concussion reported in the results?

If ethics would allow, and information were available from transcripts, greater participant information would be valuable in table 1, such as playing level (amateur is still quite broad, club/ university/ urban/ rural/ men’s/ women’s cultures can be different ), playing experience etc. A graphic explaining each ps’ characteristics would be a nice visual of who the sample represents. Reference to timelines should be added – are the Ps current players, how long ago were the concussions? This is important given that attitudes in sport change with time, as much detail re timings of playing/ concussion should be given. The limitations section mentions that one of the sample has had 20 concussions – this is very important, but isn’t referenced in the findings or participant characteristics.

Data analysis

What measures were undertaken to maintain the rigor/ quality of the analysis? Braun and Clarke have got a quality checklists (published 2019 off the top of my head) that could be applied and referenced here.

Findings

I think the results section could be expanded to further described the nuance of each situation and whether experiences were shared or diverse amongst ps with different characteristics. Greater presentation of participant quotes in-text/ in a supporting table would be beneficial- Are all ps represented meaningfully in the findings?

Are all relevant topics listed in the interview description presented in the findings? – Did concussion history/competitiveness affect attitudes? Were participant perspectives different, or consistent amongst each other.

A thematic map with example quotes, or a pen profile (e.g https://link.springer.com/article/10.1186/1471-2458-11-831) would benefit the results section. Could any links be made between knowledge and attitudes – does one preceed/influence the other?

Discussion

I think the discussion could be strengthened with greater references to broader literature, and discussion of the study’s results/ players experiences, rather than references essentially to the theme titles.

“For one to state that they are aware of potential neurodegenerative diseases and to choose to continue participating in Rugby, would lead one to believe they do not fully understand the risk, and in-fact have poor knowledge or negative concussion attitude.”- This reads in quite a sensationalist manner - arguably there is a big jump between concussion awareness/ appropriate management and awareness of the link between neurodegenerative disease here. This point could be supported by work that highlights that concussion knowledge doesn’t always lead to attitude change, and more recent literature generally could be cited in this chapter. Does the Tadmor paper on 278 state why players would dissuade family members or describe the concussion knowledge of the sample and link it to this point? In the current writing I’m not convinced that the Tadmor reference is able to support the previous sentence.

This discussion could present a more critical references to the literature, there is a fair amount of research that discusses responsibility in concussion and the bystander effect which would add value here. Gender differences have been cited in concussion experience/ management, are your findings consistent with that literature?`. Some sweeping statements are made without link to references, or acknowledgement as grey/anecdotal sources.

Line 283 – this paragraph could be expanded with more detail. Is the bystander effect relevant here, there are some papers (by Danielle salmon I think) that reference this in rugby.

295 colloquial language here, try to avoid. Are there any references/ theories that could back up this next sentence? The authors interpret that poor duty of care is due to lack of education, but did your sample offer their perspectives on this?

317 this reads as an opinion, I think there would be more value describing the current landscape of attitudes, rather than what should happen in an ideal world. Or instead, add a reference to some coach/ referee policy/code of conduct that references responsibility/duty of care.

Line 327 this paragraph could be supported by wider references. The Chris nowinski bradford hill paper would be relevant here as would a further explanation of the point made in lines 329-331. Can you quantify/explain further how the Walker 23 paper “partially rectifies this” ?

In 336 I am struggling to understand what comparison is being made. 341-43 is a sweeping statement, could existing literature that discusses access to concussion information be discussed instead – just because the documents are passively available doesn’t mean players make efforts to access them!

343 – colloquial/sensationalist language used. Could these media articles be referenced? Broader reference to the representation of concussion in the media could be made, or the the capacity of the media to influence behaviour?

357 – I am not sure of the concept of full education, could this be worded differently. 358 Can this be backed up with theory/references to strengthen it?

Is the general lack of pitch-side healthcare in amateur sport relevant here / in the discussion generally- ?

Limitations

A small rich sample of data is a strength here rather than a limitation, and adds missing richness to often quantitative investigations of this topic. Would suggest framing as such!

Its surprising that more ps details are offered for the first time in the limitations section – particularly for the ps with over 20, this information would be very useful referenced in the ps characteristics/ results.

382 begins as a limitation, but as the paragraph develops, I feel it could be incorporated in the discussion instead.

Conclusion

401- I think this should be reframed, what would be a “full” education? Awareness of current guidelines instead. This conclusion seems too broad given the sample included 9 players, yet it speaks for UK amateur players as a whole.

This paper has great potential. I hope the comments provided will strengthen this work

6. PLOS authors have the option to publish the peer review history of their article (what does this mean? ). If published, this will include your full peer review and any attached files.

**Do you want your identity to be public for this peer review?** For information about this choice, including consent withdrawal, please see our Privacy Policy .

Reviewer #1: No

Reviewer #2: No

---

## [Author Response · Author response to Decision Letter 1]

28 Mar 2025

Please refer to responses and amendments document attached.

Kind regards,

Dr Walker

---

## [Decision Letter · Decision Letter 1]

8 May 2025

PONE-D-24-58092R1“I’ve not been knocked out, so I’ll probably be fine.” Amateur Rugby players do not know the risks they are taking: This needs to changePLOS ONE

Dear Dr. Walker,

Thank you for submitting your manuscript to PLOS ONE. After careful consideration, we feel that it has merit but requires minor amendments to fully meet PLOS ONE’s publication criteria. Therefore, we invite you to submit a revised version of the manuscript that addresses the points raised during the review process. One of the reviewers is happy for this work to be accepted and published, whilst the other reviewer wishes you to provide further clarification of how bias in selection of participants was controlled. They also suggest altering the title to a simpler one. My recommendation as Academic Editor on this would be to remove the "this needs to change" part of the title.

We look forward to receiving your revised manuscript.

Kind regards,

Christopher Kirk, PhD

Academic Editor

PLOS ONE

Journal Requirements:

Reviewers' comments:

Reviewer's Responses to Questions

**Comments to the Author**

1. If the authors have adequately addressed your comments raised in a previous round of review and you feel that this manuscript is now acceptable for publication, you may indicate that here to bypass the “Comments to the Author” section, enter your conflict of interest statement in the “Confidential to Editor” section, and submit your "Accept" recommendation.

Reviewer #2: All comments have been addressed

Reviewer #3: (No Response)

2. Is the manuscript technically sound, and do the data support the conclusions?

Reviewer #2: Yes

Reviewer #3: Partly

3. Has the statistical analysis been performed appropriately and rigorously? 

Reviewer #2: Yes

Reviewer #3: No

4. Have the authors made all data underlying the findings in their manuscript fully available?

Reviewer #2: Yes

Reviewer #3: Yes

5. Is the manuscript presented in an intelligible fashion and written in standard English?

Reviewer #2: Yes

Reviewer #3: Yes

6. Review Comments to the Author

Reviewer #2: (No Response)

Reviewer #3: The authors conducted qualitative research of awareness and attitude towards concussion on 9 amateur rugby players. Through reflexive thematic analysis, “Poor duty of care” and “lack of education” were found to be the main reason for encouraging continued participation from the player with suspected concussion. The findings were potentially important considering the high prevalence of concussion in British amateur Rugby .

One worry about the findings is that the conclusions could be biased from several aspects. The subjects could volunteer to participate in the study for certain biased reasons. Even though qualitative research is less influenced by numbers, the low number of subjects makes it difficult to conclude about “prevalence” or generalize the findings. Also, the study is based on retrospective and self-reported data by nature.

The article title seems lengthy and unusual.

7. PLOS authors have the option to publish the peer review history of their article (what does this mean? ). If published, this will include your full peer review and any attached files.

**Do you want your identity to be public for this peer review?** For information about this choice, including consent withdrawal, please see our Privacy Policy .

Reviewer #2: No

Reviewer #3: No

---

## [Author Response · Author response to Decision Letter 2]

9 May 2025

Please see responses and amendments document attached.

---

## [Editor Report · Decision Letter 2]

12 May 2025

“I’ve not been knocked out, so I’ll probably be fine.” Amateur Rugby players do not know the risks they are taking

PONE-D-24-58092R2

Dear Dr. Walker,

We’re pleased to inform you that your manuscript has been judged scientifically suitable for publication and will be formally accepted for publication once it meets all outstanding technical requirements.

Kind regards,

Christopher Kirk, PhD

Academic Editor

PLOS ONE

---

## [Editor Report · Acceptance letter]

PONE-D-24-58092R2

PLOS ONE

Dear Dr. Walker,

I'm pleased to inform you that your manuscript has been deemed suitable for publication in PLOS ONE. Congratulations! Your manuscript is now being handed over to our production team.

Kind regards,

on behalf of

Dr. Christopher Kirk

Academic Editor

PLOS ONE